# Mark3DGS: Protecting the Intellectual Property of 3D Gaussian Splatting with Robust Watermarking

## Abstract

3D Gaussian Splatting (3DGS) has become a leading technique in computer vision and graphics, offering photorealistic scene representation and real-time rendering. However, due to high computational demands and the sensitivity of training data, 3DGS models face significant intellectual property theft risks, yet current protection mechanisms are insufficient. In this paper, we introduce Mark3DGS, a novel watermarking framework designed to protect 3DGS models. The framework includes perception-aware pruning for efficient Gaussian reduction, uncertainty-frequency-guided HVQ for resilient watermark embedding, tile-based rasterization with early termination and caching for optimized splatting, and adaptive extraction strategies for reliable watermark recovery. Additionally, we present MarkGS-Sim, a platform to evaluate watermark robustness across various 3DGS variants and conditions. Experimental results show that Mark3DGS outperforms state-of-the-art methods in watermark capacity, imperceptibility, and computational efficiency, achieving 206 FPS rendering, minimal storage ($< 200$MB), compatibility with multiple 3DGS variants, and strong robustness to various watermark attacks.

3D Gaussian Splatting (3DGS) Charatan et al. (2024) has emerged as a preeminent technique in computer vision and graphics, distinguished by its exceptional accuracy, accelerated rendering speed, and superior generalization capabilities Li et al. (2024b). Specifically, 3DGS reconstructs 3D scenes as Gaussian primitives with parameters like color, opacity and scale, optimized iteratively for novel view synthesis. Given the high computational cost and potential use of private data in training 3DGS models, their theft or misuse poses serious security risks Fei & Xu (2024). Thus, safeguarding the intellectual property (IP) of 3DGS models is crucial, yet existing research remains limited.

Digital watermarking techniques Wang et al. (2024); Zhu et al. (2024) have proven effective for IP protection across media like images Xu et al. (2024), videos Gull & Parah (2024), and 3D content Liu et al. (2024a), evaluated by capacity (embedded bits), invisibility (visual impact), and robustness (resistance to attacks like compression or cropping). Intuitive approaches apply 2D watermarking methods such as HiDDeN Zhu et al. (2018) or MBRS Jia et al. (2021) to rendered images, but they fail to ensure reliable message extraction and only protect the images, not the underlying 3D model. Malicious users could steal the model and rerender manipulated content, undermining integrity. Traditional explicit protections embed watermarks into point

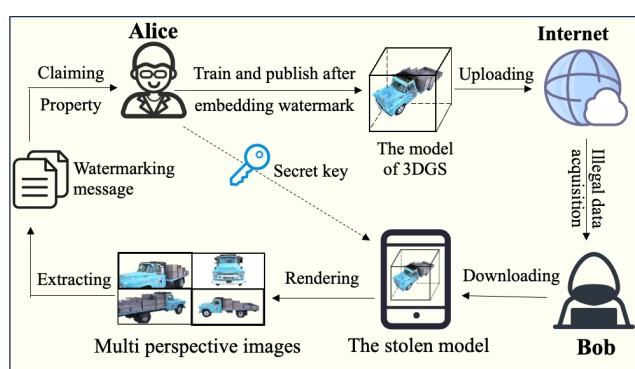

Figure 1: Application scenario process of Mark3DGS. In the event that 3DGS models are stolen by malicious user (Bob), Mark3DGS enables Alice to claim ownership of the model by transmitting watermarking messages embedded within the models to the rendered new perspective images.

clouds or meshes via wavelet transforms Wang et al. (2024), while recent NeRF-based methods

Luo et al. (2025) modify weights or color functions. However, these are ill-suited for 3DGS's explicit representation, where primitive attributes have physical meanings—direct embedding risks fidelity loss—and the vast number of primitives inflates memory/computation, limiting capacity and robustness.

To address these challenges Amrit & Singh (2022), we propose Mark3DGS, a novel secure framework for 3DGS IP protection, with the application scenario illustrated in Figure 1. It prunes low-impact primitives via perception-aware scoring and adaptive thresholds, compresses SH coefficients with uncertainty- and frequency-guided HVQ, embeds watermarks using SVD modifications and distortion layers, optimizes rendering via tile-based rasterization with early termination and caching, and extracts messages adaptively with uncertainty/frequency refinement and residual compensation. The primary contributions of this paper are as follows:

- A novel intellectual property protection scheme, Mark3DGS, is proposed, offering a robust watermarking solution for 3DGS models with broad applications in 3D asset privacy protection, encrypted communication, and secure automatic driving.

- Technically, Mark3DGS first prunes low-impact-score primitives and merges proximate points to reduce storage while preserving fidelity; it then applies uncertainty- and frequency-guided HVQ for SH coefficient compression, followed by SVD-based watermark embedding with distortion layers for robustness and invisibility; finally, tile-based rasterization enables parallel splatting with early termination and local caching for efficient rendering.

- A universal simulator for Mark3DGS, termed MarkGS-Sim, is proposed to deliver a cohesive simulation-rendering pipeline for evaluating watermark robustness under dynamic operations, physical simulations, and across various 3DGS variants.

- Comparative experiments demonstrate that Mark3DGS surpasses state-of-the-art methods in watermark capacity, imperceptibility, and computational efficiency, with seamless extension to diverse 3DGS variants and exceptional robustness against attacks like compression, noise, and model manipulations.

## 1 RELATED WORK

### 1.1 3D GAUSSIAN SPLATTING

Traditional 3D reconstruction methods Huang et al. (2024b) combine environmental cues and camera images but face limitations like poor performance with occluded objects and slow real-time processing. To enable real-time 3D reconstruction, 3D Gaussian Splatting has emerged as a more efficient approach. The original work Kerbl et al. (2023) used Gaussian primitives to generate 3D models through iterative optimization. Using a renderer, the model's geometry and texture are combined to produce realistic images based on the viewpoint and lighting. Recently, variants of 3DGS Huang et al. (2024a); Li et al. (2024a); Deng et al. (2024); Tang et al. (2023) have led to applications in autonomous driving Zhou et al. (2024b), surgical reconstruction Liu et al. (2024b), and 3D content generation Tang et al. (2023). Mip-splatting Yu et al. (2024) addressed high-frequency artifacts caused by sampling rate optimization, while Octree-GS Ren et al. (2024) improved reflection details with shadow optimization. Endo-4DGS Huang et al. (2024d) utilized 3DGS for real-time dynamic scene rendering, overcoming pose inaccuracies.Despite its advantages, training 3DGS models remains challenging and they often require private data, making intellectual property protection crucial Zhou et al. (2024a).

### 1.2 3DGS WATERMARKING

Several recent studies have explored the integration of watermarking into 3DGS models. Specifically, Zhang et al. introduced GS-Hider Zhang et al. (2024), a steganographic framework that embeds 3D scenes and images into Gaussian primitive clouds using coupled secure features and dual decoders to disentangle RGB scenes from hidden messages. Jang et al. proposed 3D-GSW Jang et al. (2025), leveraging Discrete Fourier Transform (DFT) to embed binary messages into high-frequency regions, enabling imperceptible watermarking. Chen et al. presented GuardSplat Chen et al. (2025), a CLIP-guided pipeline designed for efficient watermarking, while Ren et al. Ren

et al. (2025) developed a key-secured scheme through joint optimization of 3DGS and a dedicated decoder. To enhance fidelity and security, Zhang et al. Zhang et al. (2025) introduced anchor points and neural decoding. To preserve rendering quality, In et al. proposed CompMarkGS In et al. (2025), which optimizes image embedding within 3DGS assets. Huang et al. introduced GaussianMarker Li et al. (2025a), applying uncertainty-aware perturbations to Gaussian parameters to enhance robustness against distortion. For generative Gaussian models, Li et al. proposed GaussianSeal Li et al. (2025b), focusing on watermark embedding during generation. Despite these advancements, existing methods suffer from limitations such as long training times, high memory usage, dependency on large pre-trained modules, and increased system complexity Charatan et al. (2024). These factors hinder real-time deployment on memory-constrained devices and limit scalability. In response, we propose a novel watermarking framework for 3DGS that efficiently embeds information into compressed spherical harmonics coefficients while preserving high rendering fidelity and ensuring strong robustness against various watermark attacks.

## 2 PRELIMINARIES: REPRESENTATION OF 3DGS

In the 3DGS method, reconstructed 3D model is represented by millions of Gaussian primitives, each defined by attributes like position, scaling, rotation coefficients, opacity, and SH coefficients for view-dependent color. These points render new images through anisotropic volumetric "splatting" onto a 2D plane. The 3DGS is initialized using sparse point clouds from Structure from Motion (SFM) Kerbl et al. (2023), and each Gaussian primitive is expressed as:

$$G(\boldsymbol{x}) = e^{-\frac{1}{2}\boldsymbol{x}^T \Sigma^{-1} \boldsymbol{x}}, \tag{1}$$

where $\boldsymbol{x}$ denotes the mean 3D position vector, $\Sigma$ is the the covariance matrix formulated using a 3D scale vector $S$ and a rotation matrix $R$, $\Sigma'$ denotes the corresponding covariance matrix projection, which can be defined as

$$\Sigma = RSS^T R^T, \Sigma' = JP\Sigma P^T J^T. \tag{2}$$

where $P$ is the world-to-camera matrix and $J$ is the Jacobian of the affine approximation of the projective transform Huang et al. (2024b). For each Gaussian primitive, the axis-aligned bounding box encapsulating the 99% confidence ellipse of each 2D projected covariance is determined. Next, the color of a pixel $C$ is calculated using $\mathcal{N}$ Gaussian primitives that intersect the pixel. These primitives are then sorted by their depth values and blended accordingly. i.e.,

$$C = \sum_{i \in \mathcal{N}} c_i \alpha_i \prod_{j=1}^{i-1} (1 - \alpha_j), \tag{3}$$

where $c_i, \alpha_i$ denotes the view-dependent color and opacity Yu et al. (2024), derived from a 2D Gaussian with covariance $\Sigma$, multiplied by an optimizable per-3D Gaussian's opacity Huang et al. (2024d). Therefore, each Gaussian primitive is characterized by the following Gaussian parameters: position $\mathbf{P} \in \mathbb{R}^3$, color defined by SH coefficients $\mathrm{SH} \in \mathbb{R}^{(\kappa+1)^2} \times 3$ (where $\kappa$ is the degrees of freedom), opacity $\alpha \in \mathbb{R}$, rotation factor $\mathbf{R} \in \mathbb{R}^4$, and scaling factor $\mathbf{S} \in \mathbb{R}^3$ Liu et al. (2024b).

## 3 PROPOSED METHOD

### 3.1 GAUSSIAN PRIMITIVES PRUNING

The original 3DGS achieves high-fidelity rendering through dense Gaussian distributions, but this results in significant storage overhead and reduces watermarking efficiency. To address this, we propose a perception-aware pruning method that combines neural network pruning principles with recent 3DGS compression techniques. This method selectively removes low-impact primitives, improving storage efficiency and enabling more effective watermark embedding.

We introduce an impact score metric that evaluates each Gaussian based on its geometric, textural, and spatial attributes, quantifying its contribution to the rendered image. Only visually significant primitives are retained. To enhance score accuracy, we incorporate Discrete Wavelet Transform (DWT) Jang et al. (2025) to analyze high- and low-frequency content, overcoming limitations in

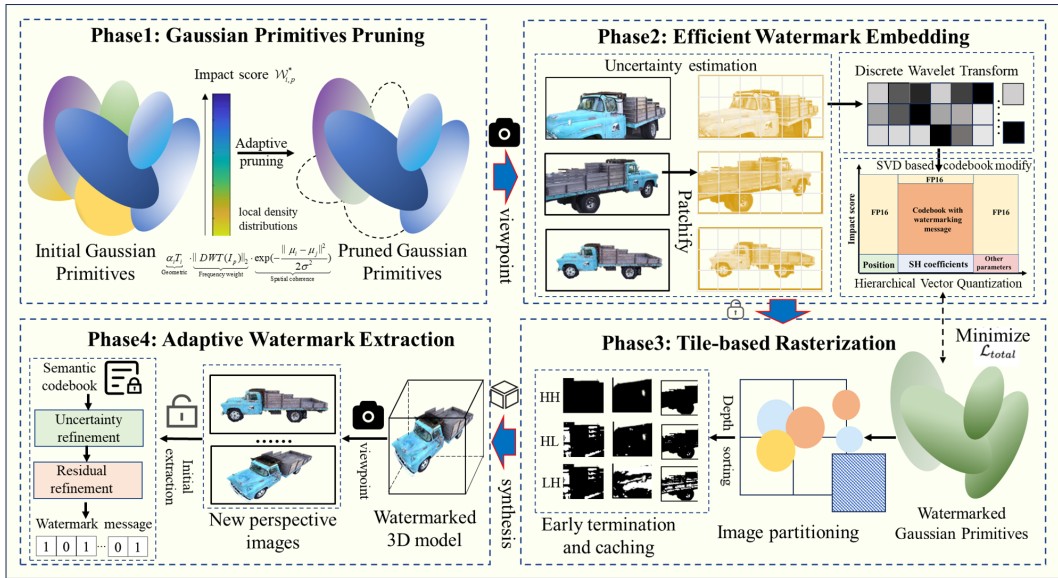

Figure 2: The pipeline of the Mark3DGS begins with perception-aware Gaussian primitives pruning using DWT frequency analysis for impact scores and adaptive low-contribution removal to reduce storage while preserving fidelity. Laplace uncertainty estimation prioritizes SH coefficients for frequency-guided multi-stage HVQ compression, SVD watermark embedding with distortion layers for robustness, followed by tile-based rasterization for parallel splatting with early termination and caching. Adaptive extraction reverses HVQ through frequency weighting and residual compensation for robust recovery.

texture-less regions. The impact score is calculated as:

$$\mathcal{W}^*_{i,p} = \underbrace{\alpha_i T_i}_{\text{Geometric}} \cdot \underbrace{\|DWT(I_p)\|_2}_{\text{Frequency weight}} \cdot \underbrace{\exp(-\frac{\|\mu_i - \mu_j\|^2}{2\sigma^2})}_{\text{Spatial coherence}}, \tag{4}$$

where $DWT(\cdot)$ computes discrete wavelet coefficients to capture frequency content at pixel $p$, $\mu_j$ denotes the position of the nearest neighbor Gaussian primitives, $\alpha_i$ represents the opacity contribution of the $i$-th Gaussian, $T_i$ is its geometric influence factor, and $\sigma$ defines the spatial coherence scale influencing the exponential decay based on the Euclidean distance $\|\mu_i - \mu_j\|$ between the $i$-th Gaussian and its neighbor. Next, we implement an adaptive pruning strategy that dynamically adjusts compression ratios based on local density distributions. Unlike fixed-threshold methods Chen et al. (2025), our approach adapts to scene-specific characteristics:

$$\tau = \tau_{base} + \gamma \cdot \frac{1}{N_k} \sum_{k=1}^{K} \mathbb{I}(\mathcal{W}^*_k < \tau_{base}), \tag{5}$$

where $\gamma$ controls pruning aggressiveness by scaling the adaptive adjustment, $N_k$ is the number of Gaussians in the $k$-th cluster, $\tau_{base}$ represents the baseline threshold for initial pruning, and $\mathbb{I}(\mathcal{W}^*_k < \tau_{base})$ is an indicator function that equals 1 if the impact score $\mathcal{W}^*_k$ of the $k$-th cluster falls below $\tau_{base}$, otherwise 0, driving the dynamic threshold $\tau$ based on local density distributions. This ensures locally optimized thresholds for better compression without sacrificing visual quality Amrit & Singh (2022). After adaptive pruning, we apply refined clustering to merge spatially proximate Gaussian primitives using an anisotropic distance metric.

## 3.2 EFFICIENT WATERMARK EMBEDDING

After pruning low-impact Gaussian primitives, the remaining high-contribution Gaussians primarily consist of SH coefficients, which dominate storage. Direct embedding into these coefficients can degrade geometric and visual fidelity. To address this, we enhance the Hierarchical Vector

Quantization (HVQ) technique with uncertainty-aware and frequency-guided mechanisms. This hybrid approach compresses SH coefficients into a compact semantic codebook, by prioritizing high-uncertainty and frequency-rich regions. Uncertainty estimation via Laplace approximation prioritizes coefficients above $\tau_{unc}$ for quantization. Next, we apply frequency-guided densification using DWT to identify texture-rich regions, where watermarks are less perceptible. The impact score from pruning is augmented with frequency weights: $\mathcal{W}_i^{freq} = \mathcal{W}_i^* \cdot \|DWT(\mathbf{g}_i)\|_2$.

For watermark embedding, we distribute the binary message $M_L$ across stages by modifying codebook indices via singular value decomposition (SVD), ensuring imperceptibility as $\mathbf{c}_k' = \mathbf{c}_k + \Delta\mathbf{c}_k$, where $\Delta\mathbf{c}_k$ encodes watermark bits with minimal norm. To enhance robustness, we simulate common attacks (e.g., JPEG compression) with differentiable distortion layers during fine-tuning. We further optimize the watermark embedding by minimizing the cumulative embedding distortion $\mathcal{D}$:

$$\mathcal{D} = \sum_{k=1}^{K} \sum_{j \in \mathcal{R}(\mathbf{c}_k)} \|\mathbf{g}_j - \mathbf{c}_k'\|_2^2 + \lambda \cdot \sum_{k=1}^{K} \|\mathbf{c}_k' - \mathbf{c}_k\|_2^2, \tag{6}$$

where $\lambda$ balances the fidelity between quantized and watermarked coefficients by weighting the deviation penalty, $\mathcal{R}(\mathbf{c}_k)$ denotes the set of Gaussian indices associated with the $k$-th code vector, $\mathbf{g}_j$ represents the original SH coefficient vector for the $j$-th Gaussian, $\mathbf{c}_k'$ is the watermarked code vector, and $\mathbf{c}_k$ is the original quantized code vector, ensuring a trade-off between watermark robustness and visual quality. Finally, the residual error is quantified as:

$$\mathcal{E}_k = \sum_{i=1}^{d} (\mathbf{g}_i - \mathbf{c}_k')^2 + \lambda_{freq} \sum \|DWT(\mathbf{g}_i - \mathbf{c}_k')\|_{sub}^2, \tag{7}$$

where $\lambda_{freq}$ is the frequency weighting parameter that balances the high-frequency subband penalties, $d$ is the dimension of the SH coefficients, and the summation applies DWT to the residuals over subbands ($sub \in \{LH, HL, HH\}$) to penalize distortions in texture-rich components.

### 3.3 TILE-BASED RASTERIZATION

To enhance watermark embedding and rendering efficiency, we introduce a tile-based rasterization framework inspired by high-performance graphics systems Liu et al. (2024b). The rendered image is divided into fixed-size tiles ($16 \times 16$ pixels), where localized, parallelized operations for Gaussian splatting, watermark integration, and uncertainty/frequency computations are performed. This reduces computational overhead and ensures smooth integration of distortion simulations and loss optimization. Each tile $T_i$ processes Gaussian primitives based on their depth and impact scores $\mathcal{W}_i^{freq}$, augmented with uncertainty $U_i$. If the accumulated opacity for a pixel reaches a saturation threshold $\tau$, processing terminates early to avoid redundant computations:

$$\text{If } \sum_{j=1}^{M} \alpha_j(p) \cdot \mathcal{W}_j^{freq} \geq \tau \Rightarrow \text{terminate } p \text{ processing}. \tag{8}$$

A tile-local caching mechanism is implemented to reduce memory bottlenecks, storing intermediate fragments before committing them to the global frame buffer. The pipeline optimization is guided by the joint loss from the embedding stage, adapted for perceptual and watermark fidelity:

$$\mathcal{L}_{\text{total}} = \lambda_{msg}\mathcal{L}_{msg} + \lambda_{rec}\mathcal{L}_{rec} + \lambda_{wavelet}\mathcal{L}_{wavelet}, \tag{9}$$

where $\mathcal{L}_{msg}$ is binary cross-entropy, $\mathcal{L}_{rec}$ is MAE reconstruction, and $\mathcal{L}_{wavelet}$ penalizes high-frequency distortions. The Gaussian parameters $\theta$ are updated using Adam optimization: $\theta_{t+1} = \theta_t - \eta \cdot \nabla\mathcal{L}_{\text{tile}}$, with learning rate $\eta$ adjusted by tile-specific uncertainty to refine embedding without compromising pruning efficiency.

### 3.4 ADAPTIVE WATERMARK EXTRACTION

During the watermark extraction stage, the embedded watermark information is efficiently retrieved by reversing the enhanced HVQ operations applied during uncertainty- and frequency-guided quantization of SH coefficients. Specifically, the watermark is extracted from modified codebook indices

by decoding the semantic codebook structure, leveraging prioritization of high-uncertainty and high-frequency regions from embedding. This enables robust recovery without complex iterative decoding, ensuring efficiency in distorted scenarios. Initially, we extract the watermark bit sequence from watermarked codebook vectors $\mathbf{C}' = \{\mathbf{c}'_1, \mathbf{c}'_2, \ldots, \mathbf{c}'_K\}$ using inverse SVD:

$$w_i = \text{round}\left(\frac{\|\mathbf{c}'_k - \mathbf{c}_k\|_2}{\beta \cdot \delta}\right) \cdot I\left(\|\mathbf{c}'_k - \mathbf{c}_k\|_2 \leq \epsilon\right), \tag{10}$$

where $\beta$ is the uncertainty-based embedding strength, $\delta$ is a global scaling parameter that controls the overall embedding strength, and $\epsilon$ serves as a tolerance threshold for noise filtering to prevent false positives.

To enhance robustness, we apply uncertainty-guided refinement, weighting bits by SH coefficient uncertainty $U_i$ to prioritize high-uncertainty regions and minimize errors under distortions like noise or compression. Additionally, frequency-domain reliability via DWT on code vector differences refines extraction, combining scores with uncertainty weighting to favor bits in less perceptible high-frequency areas, bolstering resilience against simulated attacks. Finally, multi-stage HVQ residuals enable iterative refinement by recalculating frequency-weighted residual error:

$$r_k = \sum_{i=1}^{d}(\mathbf{g}_i - \mathbf{c}'_k)^2 + \lambda_{freq} \sum \|DWT(\mathbf{g}_i - \mathbf{c}'_k)\|^2_{sub}, \tag{11}$$

compensating for quantization deviations and aligning with cumulative distortion minimization in equation 9. Through these adaptive strategies integrated with embedding's joint loss, our method achieves high accuracy and robustness, ideal for real-time, memory-constrained devices.

## 4 THE UNIVERSAL SIMULATOR FOR MARK3DGS

To evaluate Mark3DGS's robustness in realistic conditions, we propose MarkGS-Sim, a unified simulation-rendering platform for watermarked 3DGS models. As shown in Figure 3, it enables dynamic operations (addition, deletion, editing) and supports physical/environmental simulations like deformation, collision, and view distortions, with compatibility for 3DGS variants to facilitate comprehensive evaluations. The pipeline reconstructs scenes from multi-sensor inputs (LiDAR, camera, dataset) via LIO and COLMAP, converting point clouds into Gaussian kernels with visual (SH coefficients), physical (velocity, strain), and watermark attributes. For physical modeling, we apply position-based dynamics and compute deformation gradients:

$$\mathbf{F} = [\mathbf{x}_1 - \mathbf{x}_0, \mathbf{x}_2 - \mathbf{x}_0, \mathbf{x}_3 - \mathbf{x}_0] \cdot [\mathbf{x}_1^0 - \mathbf{x}_0^0, \mathbf{x}_2^0 - \mathbf{x}_0^0, \mathbf{x}_3^0 - \mathbf{x}_0^0]^{-1} \tag{12}$$

where $\mathbf{F}$ is the deformation gradient matrix for modeling strain in physical interactions, with $\mathbf{x}_0$ to $\mathbf{x}_3$ as current deformed tetrahedron vertices in the Gaussian kernel and $\mathbf{x}_0^0$ to $\mathbf{x}_3^0$ as initial rest positions, the inverse ensuring accurate covariance and SH updates during deformations.

Scene construction supports segmenting and inpainting watermarked models via 2D-to-3D masks, adding new objects through hierarchical embedding in bounding volumes for seamless integration. Editing like scaling and rotation updates kernel properties while tracking SH changes as $C'_k = C_k + \Delta C_k$, where $\Delta C_k$ reflects interaction modifications, with watermark extraction verified via differential SH analysis. Optimizations such as CUDA acceleration, LOD control, tile-based rasterization, and float16 precision reduce latency and memory usage. By modularizing Gaussian kernels, the simula-

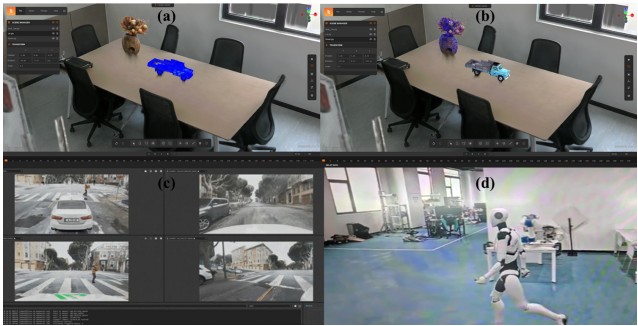

Figure 3: The function demonstration of MarkGS-Sim. We can edit watermarked 3D objects and extract watermarking in the scene as subfigures (a) and (b), demonstrating scalability for 4DGS (c) and DreamGS (d).

tor generalizes to 3DGS variants—e.g., temporal terms for 4DGS or compression pruning for DreamGS—enabling hybrid scenes. MarkGS-Sim thus serves as a versatile tool for robust watermark verification of our method under real-world scene.

Table 1: Comparison of quantitative results under different message lengths.

| Methods | 16 bit | | | 32 bit | | | 48 bit | | |
|---|---|---|---|---|---|---|---|---|---|
| | Bit Acc. | PSNR | LPIPS | Bit Acc. | PSNR | LPIPS | Bit Acc. | PSNR | LPIPS |
| Hidden+3DGS | 63.07% | 25.59 | 0.0471 | 58.25% | 25.31 | 0.0489 | 53.02% | 21.42 | 0.0533 |
| GS-hider | 94.89% | 26.86 | 0.0385 | 92.35% | 25.77 | 0.0411 | 91.43% | 22.84 | 0.0483 |
| WATER-GS | 94.23% | 25.71 | 0.0319 | 93.75% | 24.83 | 0.0414 | 93.03% | 23.94 | 0.0545 |
| GaussianMarker | 95.12% | 26.43 | 0.0305 | 93.08% | 24.67 | 0.0377 | 89.39% | 22.73 | 0.0564 |
| GuardSplat | 96.64% | 28.55 | 0.0312 | 94.04% | 28.41 | 0.0339 | 93.29% | 27.90 | 0.0378 |
| **Mark3DGS** | **99.45%** | **29.69** | **0.0117** | **98.79%** | **28.52** | **0.0209** | **98.06%** | **28.16** | **0.0233** |

## 5 EXPERIMENT AND NUMERICAL RESULTS

### 5.1 EXPERIMENTS SETTING

**Dataset and Baseline.** We evaluate Mark3DGS using widely adopted 3DGS datasets, including Tanks & Temples Knapitsch et al. (2017), MipNeRF360 Barron et al. (2022), and the Blender dataset Collins et al. (2022), as well as our custom Fyts dataset, which contains 180 real-world images of large-scale architectural structures. Experiments cover small, medium, and large-scale scenes: MipNeRF-360 (Vase) and Blender (Desk), Tanks & Temples (Truck), and Fyts (Stadium). We compare Mark3DGS with five SOTA methods: HiDDeN Zhu et al. (2018) + 3DGS Kerbl et al. (2023), GS-hider Zhang et al. (2024), WATER-GS Tan et al. (2024), GaussianMarker Huang et al. (2024c), and GuardSplat Chen et al. (2025). Note that the qualitative results in this section are the average of the above four datasets.

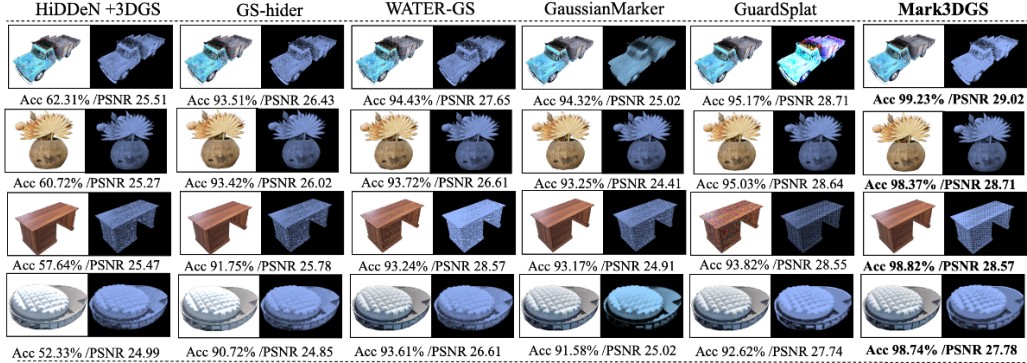

Figure 4: Visual quality comparison with baselines, results are shown for 32-bit watermarking messages. For each method, the original rendering (left) and its difference map (right) are displayed.

**Implementation details.** We conducted a series of simulations on a server equipped with an Intel Xeon CPU and an NVIDIA GTX 4090 GPU, utilizing the MarkGS-Sim to implement Mark3DGS. We use the key metrics from Tan et al. (2024) to evaluate the watermark performance. When pruning Gaussian primitives phase, we set the $\gamma = 0.2$ for adaptive threshold. For uncertainty estimation in watermark embedding, we use the Laplace approximation with threshold $\tau_{unc} = \bar{U} + \sigma_U$, and set the codebook adaptation decay $\lambda_d = 0.8$ in multi-stage HVQ. In tile-based rasterization, we apply an opacity saturation threshold $\tau = 0.95$ for early termination. Fine-tuning employs the Adam optimizer over 10,000 iterations with joint loss weights $\lambda_{msg} = 0.4$, $\lambda_{rec} = 1.0$, and $\lambda_{wavelet} = 0.3$.

### 5.2 CAPACITY AND IMPERCEPTIBILITY

**Quantitative Results.** We conducted three experiments for each watermarking message length, as shown in Table 1, illustrating the relationship between bit accuracy and message length. As expected, bit accuracy decreases with longer messages, a common trade-off in watermarking techniques. However, Mark3DGS consistently outperforms other methods, maintaining high bit accuracy even with longer payloads. Additionally, high PSNR and low LPIPS values show that our

Table 2: Comparisons of the state-of-the-art methods for bit accuracy w.r.t various distortion types. We show the results on 32-bit watermarking messages.

| Methods | No Distortion | Rotation $(\pm\pi/6)$ | Scaling $(\leq 25\%)$ | Crop $(40\%)$ | JPEG $(10\%)$ | Gaussian Noise $(\mu = 0.1)$ | Cloning $(50\%)$ | Pruning $(20\%)$ |
|---|---|---|---|---|---|---|---|---|
| HiDDeN+3DGS | 58.25 | 51.74 | 55.25 | 47.62 | 50.71 | 53.36 | 54.21 | 46.63 |
| GS-hider | 92.35 | 91.98 | 87.68 | 85.52 | 88.79 | 90.54 | 86.43 | 85.52 |
| WATER-GS | 95.18 | 92.67 | 93.15 | 92.70 | 89.42 | 88.62 | 91.43 | 90.22 |
| GaussianMarker | 93.75 | 92.42 | 91.34 | 87.42 | 86.52 | 88.29 | 90.82 | 88.48 |
| GuardSplat | 94.04 | 92.41 | 93.43 | 89.45 | 87.29 | 92.18 | 92.42 | 88.51 |
| **Mark3DGS** | **98.79** | **94.51** | **93.73** | **91.59** | **92.21** | **93.55** | **91.62** | **90.27** |

method preserves visual fidelity and structural integrity. While HiDDeN+3DGS achieves high imperceptibility, it struggles with bit extraction accuracy, even with short bit lengths. Mark3DGS exhibits a gradual decline in accuracy, maintaining 98.79% for 32-bit and 98.06% for 48-bit watermarks, outperforming methods like WATER-GS and GS-hider, which show sharper degradation. This confirms Mark3DGS's superior capacity for embedding and extracting watermarks efficiently, even with larger payloads.

**Qualitative Results.** We perform a visual comparison of all baseline methods in Figure 4. While all approaches achieve similar 3D reconstruction quality, their watermark invisibility differs. Mark3DGS maintains high visual fidelity without visible artifacts, showcasing its superior ability to embed watermarks covertly without compromising reconstruction. In contrast, HiDDeN+3DGS suffers from reduced bit accuracy due to inefficient watermark embedding, affecting extraction robustness. Both WATER-GS and GaussianMarker exhibit degraded visual quality from view synthesis distortions, impairing features like specular highlights and edges.

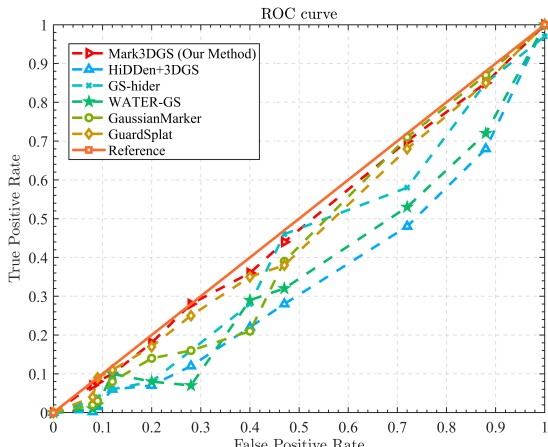

Figure 5: The ROC curve of various methods.

**Image-Level and Model-Level Attacks.** We apply several traditional watermarking attack methods as Table 2. The results show that Mark3DGS maintains high bit accuracy across all attacks, outperforming other methods. While GS-hider and WATER-GS perform similarly to Mark3DGS without distortions, their performance drops significantly under rotations, scaling, and cropping. Although post-processing reduces watermark bit accuracy, Mark3DGS exhibits minimal degradation and consistently outperforms the baselines, maintaining watermark integrity even under image-level attacks. Moreover, we evaluate the robustness of our method against model manipulation that could alter the embedded watermark. Three distortion experiments are conducted: Gaussian noise ($\mu = 0.1$), cloning (50% random duplication), and pruning (20% random removal). As illustrated in Table 2, we evaluate bit accuracy under various model attacks, with our method exhibiting superior robustness compared to existing approaches. These results substantiate that our proposed technique effectively preserves the embedded watermark, even under distortion attacks.

**Steganalysis Attack.** To evaluate the security of our Mark3DGS, we perform anti-steganography detection using StegExpose Boehm (2014) on container images generated by various methods. Each method embeds 32 bits into 3DGS objects. We vary the detection thresholds across a broad range, from 0.001 to 0.95, within StegExpose, and plot the corresponding receiver operating characteristic (ROC) curve in Figure 5. The ideal scenario assumes the detector has a 50% chance of identifying a watermark in a balanced test set, which equates to random guessing. The results clearly show that our method outperforms baseline significantly in terms of security. This demonstrates that our

| Methods | Train Time ↓ | Frames Per Second ↑ | Storage Memory ↓ |
|---|---|---|---|
| HiDDeN+3DGS | 148m16s | 62 | 411MB |
| GS-hider | 56m27s | 135 | 418MB |
| WATER-GS | 88m21s | 78 | 678MB |
| GaussianMarker | 27m52s | 117 | 358MB |
| GuardSplat | 24m15s | 64 | 498MB |
| **Mark3DGS** | **9m42s** | **206** | **193MB** |

Table 3: Computational efficiency comparison on the Tanks & Temples dataset, results are shown for 32-bit message.

| Methods | PSNR ↑ | LPIPS↓ | Bit Acc ↑ |
|---|---|---|---|
| 2DGS | 25.94 | 0.168 | N/A |
| Ours+2DGS | 24.34 | 0.178 | 93.45% |
| 4DGS | 25.89 | 0.198 | N/A |
| Ours+4DGS | 23.16 | 0.268 | 92.71% |
| Compact3D | 26.53 | 0.192 | N/A |
| Ours+Compact3D | 25.70 | 0.214 | 95.48% |
| DreamGS | 24.21 | 0.137 | N/A |
| Ours+DreamGS | 23.44 | 0.263 | 92.54% |

Table 4: Ablation studies of our method on various 3DGS frameworks.

approach is far less vulnerable to detection by steganography analysis techniques, thereby ensuring a high level of security.

## 5.3 COMPUTATIONAL EFFICIENCY

We evaluate the computational efficiency of Mark3DGS by comparing its training time, rendering speed (FPS), and storage size with baseline methods, as summarized in Table 3. WATER-GS and GuardSplat exhibit prolonged training times and low FPS, making them unsuitable for real-time applications on resource-constrained devices. GaussianMarker offers moderate performance, but Mark3DGS significantly surpasses it and all baselines with the shortest training time, highest FPS, and lowest storage. This superiority stems from perception-aware pruning reducing Gaussian primitives, uncertainty- and frequency-guided HVQ compressing SH coefficients, and tile-based rasterization with early termination and caching minimizing overhead. Compared to GS-hider, Mark3DGS achieves nearly 1.5× faster rendering and less than 50% storage, providing a highly efficient and scalable solution ideal for limited-capacity devices.

## 5.4 SCALABILITY

Mark3DGS is highly versatile and can be adapted to various 3DGS variants, including 2DGS, 4DGS, CompactGS, and DreamGaussian. As shown in Table 4, our method achieves excellent visual quality (PSNR > 23) and maintains low LPIPS (<0.270) across all variants, ensuring real-time performance. It also ensures high watermark extraction accuracy (Bit ACC > 92%) and robustness. For 2DGS, watermarks are embedded in semantic codebooks using structural similarities like opacity, position, and color. In 4DGS, watermarks are embedded in temporally stable components to ensure reliable extraction under dynamic conditions. In CompactGS, watermarks are applied after pruning, with 95.48% extraction accuracy even after compression. For DreamGaussian, our method integrates seamlessly, preserving both generation quality and watermark robustness. These results demonstrate the robustness and adaptability of Mark3DGS across various 3DGS frameworks.

## 6 CONCLUSION

In this paper, we present Mark3DGS, a memory-efficient framework for protecting the intellectual property of 3D Gaussian Splatting (3DGS) models through robust watermarking. Our approach begins with perception-aware Gaussian pruning using adaptive thresholds to eliminate low-contribution primitives. Memory-efficient watermark embedding employs uncertainty estimation and frequency guidance to compress spherical harmonics (SH) coefficients via hierarchical vector quantization (HVQ), inserting messages for robustness and strength parameters for invisibility. Tile-based rasterization optimizes rendering through partitioning for parallel splatting and caching to achieve real-time efficiency, while adaptive extraction reverses HVQ via decoding and weighting for reliable recovery under distortions. We introduce MarkGS-Sim, a universal simulator for evaluating watermark robustness across physical simulations and 3DGS variants. Experimental evaluations confirm that Mark3DGS outperforms SOTA methods in imperceptibility (PSNR >28 dB, LPIPS <0.02), computational efficiency (206 FPS rendering, <200 MB storage), broad compatibility across 3DGS variants, and superior resilience to attacks like compression and noise, paving the way for secure deployment in real-world applications.

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

## A  APPENDIX OUTLINE

This appendix is organized as follows:

- In Section B, we provide additional implementation details, including dataset, evaluation metric, baseline and implementation details.
- In Section C, we present the robustness result of our model against fine-tuning attacks.
- In Section D, we present the ablation study results, including different loss combinations, various hyper-parameter values, and the effect of pruning ratio and scaling factor.
- In Section E, we provide more detailed visual results, showcasing the scalability of our method across various 3DGS frameworks.
- In Section F, we discussed limitations of our current framework and potential directions for future work.

## B  DATASET AND MODEL DETAILS

The details of the experiments setting used in our experiments are as follows.

**Dataset.** We evaluate our method using datasets widely adopted in 3DGS research: the Tanks&Temples dataset Knapitsch et al. (2017), MipNeRF360 Barron et al. (2022), and the Blender dataset Collins et al. (2022), which comprise synthetic, bounded scenes. To assess the generalizability of the Mark3DGS framework across diverse scales, we conducted additional experiments on our custom Fyts dataset Xu et al. (2024), featuring large-scale architectural structures. The Fyts dataset consists of 180 real-world scene images captured by our collaborative robotic system. For comprehensive validation, we tested our method on datasets spanning multiple scales: 1) Small-scale: MipNeRF-360 (Vase) and Blender (Desk). 2) Medium-scale: Tanks & Temples (Truck). 3) Large-scale: Fyts (Stadium).

**Evaluation Metric.** We evaluate our methods using the following critical metrics: 1) Accuracy and Imperceptibility, which reflect the trade-off between reliable watermark extraction and minimal visual distortion. Bit Accuracy quantifies the proportion of correctly decoded watermark bits, while PSNR (Peak Signal-to-Noise Ratio), SSIM (Structural Similarity Index) and LPIPS (Learned Perceptual Image Patch Similarity) assess pixel-level fidelity and perceptual quality, respectively. 2) Robustness, measured by bit accuracy under common 3D data perturbations such as compression, noise, viewpoint changes, and geometric edits, verifies the resilience of embedded messages. 3) Computational Efficiency evaluates rendering speed (FPS), storage consumption, and training time, reflecting the practicality of our method for real-time and resource-constrained applications. 4) Scalability further assesses the adaptability of our framework across diverse 3DGS variants, confirming the generalization and extensibility of Mark3DGS in complex or dynamic environments. These metrics collectively provide a comprehensive evaluation of the method's fidelity, robustness, efficiency, and versatility.

**Baseline.** We compare our approach with five SOTA strategies to ensure a fair evaluation: 1) HiD-DeN Zhu et al. (2018) + 3DGS Kerbl et al. (2023), where the classical 2D watermarking method HiDDeN is applied to the image before training the 3DGS model; 2) GS-hider Zhang et al. (2024), a method that achieves steganography of continuous information within the 3DGS model using a scene codec; 3) WATER-GS Tan et al. (2024), a pre-trained semantic encoder-based intellectual property protection scheme; 4) GaussianMarker Huang et al. (2024c), a watermarking method for 3D Gaussian Splatting models; and 5) GuardSplat Chen et al. (2025), a robust watermarking technique for 3D models based on high-frequency Gaussian encoding. For a fair comparison, we conducted experiments under identical conditions, either using reported quantitative results or replicating experiments with official implementations.

**Implementation Details.** To evaluate the effectiveness of the proposed Mark3DGS, we conducted a series of simulations on a server equipped with an Intel Xeon CPU (2.4 GHz, 128 GB RAM) and an NVIDIA GTX 4090 GPU (80 GB SGRAM), utilizing the MarkGS-Sim to implement Mark3DGS. All comparison schemes incorporate the differential Gaussian rasterization technique of 3DGS with 5,000 steps of fine-tuning during Gaussian co-adaptation. When pruning Gaussian primitives phase, we set the $\gamma = 0.2$ for adaptive threshold. For uncertainty estimation in watermark embedding, we use the Laplace approximation with threshold $\tau_{unc} = \bar{U} + \sigma_U$, and set the codebook adaptation decay $\lambda_d = 0.8$ in multi-stage HVQ. In tile-based rasterization, we apply an opacity saturation threshold $\tau_{occ} = 0.95$ for early termination. Fine-tuning employs the Adam optimizer over 10,000 iterations with joint loss weights $\lambda_{msg} = 0.4$, $\lambda_{rec} = 1.0$, and $\lambda_{wavelet} = 0.3$. It is important to note that the quantization results presented in this section are averaged across all simulations, with each simulation repeated 10 times.

## C  FINE-TUNING ATTACKS

Model fine-tuning is a common attack strategy, where the goal is to fine-tune the watermarked Mark3DGS model to distort or remove the embedded watermark. We evaluate Mark3DGS's robustness under two attack scenarios: without clean images (w/o CI) and with clean images (w/ CI). In the w/o CI scenario, attackers use noisy images to fine-tune the model, whereas in the w/ CI scenario, they use unmodified images. Further, the attacker may or may not have access to the pose key (w/ PK or w/o PK). If the pose key is available, the attacker fine-tunes the model using specific view-points corresponding to the watermark embedding. Without the pose key, only random viewpoints can be used for fine-tuning, limiting attack precision. The results for fine-tuning attacks are shown in TABLE 5. Our experiments show that in the w/o CI scenario, fine-tuning attacks do not significantly reduce the watermark bit accuracy, regardless of the pose key access. This indicates that Mark3DGS remains robust against fine-tuning attacks without clean images. However, when both clean images and the pose key are available (w/ CI and w/ PK), the watermark accuracy decreases after several fine-tuning rounds. While the watermark accuracy does decline under these conditions, this attack scenario is difficult to achieve in practice, as it requires access to both the clean images and the pose key. This confirms that Mark3DGS is highly resilient to fine-tuning attacks, especially when the attacker lacks complete information.

Table 5: Bit accuracy under fine-tuning attacks in different settings. results are presented for 32 bits and are averaged across all instances within the dataset.

| Attack setting | 0 epochs | 100 epochs | 300 epochs | 500 epochs |
|---|---|---|---|---|
| w/o CI (w/ PK) | 99.98% | 99.98% | 99.96% | 99.92% |
| w/o CI (w/o PK) | 99.98% | 99.98% | 99.97% | 99.95% |
| w/ CI (w/ PK) | 99.98% | 70.26% | 59.46% | 53.71% |
| w/ CI (w/o PK) | 99.98% | 98.75% | 98.09% | 97.22% |

## D  ABLATION STUDY

**Different Loss Combinations.** We explore the optimal loss combinations in TABLE 6 to demonstrate how each component contributes to balancing watermark robustness and rendering quality in Mark3DGS. Compared to the original 3DGS model without any watermark-related losses (row 1),

solely optimizing the message loss $\mathcal{L}_{\text{msg}}$ (row 2) severely degrades reconstruction quality, as it introduces excessive perturbations to SH coefficients, prioritizing watermark recovery at the expense of scene fidelity. Incorporating the reconstruction loss $\mathcal{L}_{\text{rec}}$ alongside $\mathcal{L}_{\text{msg}}$ (row 3) mitigates this fidelity decline by enforcing pixel-level accuracy, yet it still yields sub-optimal results due to persistent high-frequency offsets that compromise texture details and overall visual coherence. Therefore, we further integrate the wavelet loss $\mathcal{L}_{\text{wavelet}}$ (row 4) to specifically penalize distortions in high-frequency subbands identified via DWT, effectively stabilizing SH modifications and achieving the optimal trade-off between high bit accuracy for watermark extraction and superior PSNR/SSIM for rendering quality.

Table 6: Different loss combinations with $N_L = 32$ bits watermarking message. The first row denotes the original 3DGS.

| $\mathcal{L}_{\text{msg}}$ | $\mathcal{L}_{\text{rec}}$ | $\mathcal{L}_{\text{wavelet}}$ | Bit Acc | PSNR | SSIM | LPIPS |
|---|---|---|---|---|---|---|
| | | | 58.47 | N/A | 1.0000 | 0.0000 |
| ✓ | | | 100.00 | 22.36 | 0.9327 | 0.0439 |
| ✓ | ✓ | | 99.48 | 24.47 | 0.9561 | 0.0246 |
| ✓ | ✓ | ✓ | 98.79 | 28.52 | 0.975 | 0.0209 |

**Various Hyper-parameter Values.** We investigate the sensitivity of six key hyper-parameters. As shown in Fig. 6, we vary one parameter at a time across six values, keeping the others at default. For $\tau_{\text{unc}} \in [0.8\sigma, 1.6\sigma]$, bit accuracy improves initially, peaking around $1.1\sigma$, as more uncertain coefficients are prioritized. Beyond that, performance drops due to over-permissive embedding in unstable regions. $\tau_{\text{occ}} \in [0.8, 1.6]$ affects early splatting termination; setting it to 1.2 achieves optimal trade-off between rendering fidelity and watermark density. Similarly, $\lambda_{\text{freq}} = 0.1$ is an optimal setting for preserving high-frequency textures during watermark insertion, while higher values cause underfitting. The loss weight $\lambda_{\text{msg}}$ directly affects message integrity. Bit accuracy improves steeply from 91% to 94% and then plateaus—thus we select $\lambda_{\text{msg}} = 0.2$. $\lambda_{\text{rec}}$ balances image reconstruction, where values beyond 0.5 begin to slightly reduce bit accuracy due to excessive visual fidelity optimization. Finally, $\lambda_{\text{wavelet}} = 0.05$ yields optimal bit accuracy, balancing imperceptibility in frequency subbands and embedding robustness. Based on the ablation results, we adopt the following optimal configuration for all subsequent experiments: $\tau_{\text{unc}} = 1.1\sigma$, $\tau_{\text{occ}} = 1.2$, $\lambda_{\text{freq}} = 0.1$, $\lambda_{\text{msg}} = 0.2$, $\lambda_{\text{rec}} = 0.5$, and $\lambda_{\text{wavelet}} = 0.05$.

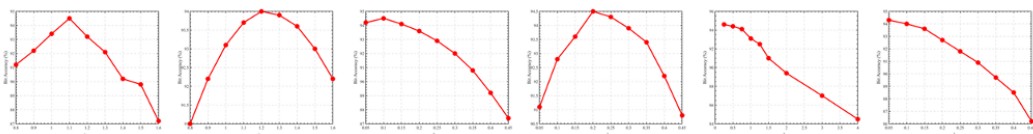

Figure 6: Performance across different hyper-parameter values with $N_L =32$ bits on the above datasets.

**The Effect of Pruning Ratio and Scaling Factor.** For adaptive pruning, our perception-aware strategy utilizes the multi-dimensional impact scores $\mathcal{W}_{i,p}^*$ to identify low-importance Gaussian primitives. Fig. 7(a) illustrates that when $\gamma \leq 0.2$, corresponding to pruning ratios around 60–70%, the rendering quality remains stable and bit accuracy slightly improves. This is attributed to the concentration of watermark embedding in visually critical regions. However, increasing $\gamma$ beyond 0.3, which induces pruning above 80%, causes a significant drop in bit accuracy due to over-pruning and loss of high-frequency details, confirming the importance of balanced pruning. In parallel, the embedding scaling factor $\zeta$ modulates the perturbation strength $\eta$ applied to SH coefficients during HVQ-based watermark embedding. As shown in Fig. 7(b), increasing $\zeta$ from 0.1 to 0.6 yields a notable improvement in bit accuracy, peaking at 98.79% with acceptable LPIPS. This indicates stronger watermark signal embedding without excessive artifact introduction. Beyond this point, larger $\zeta$ values further increasing LPIPS and decreasing bit accuracy, reflecting a trade-off between robustness and fidelity. Therefore, our analysis suggests that $\gamma = 0.2$ and $\zeta = 0.6$ provide the best compromise between imperceptibility and robustness.

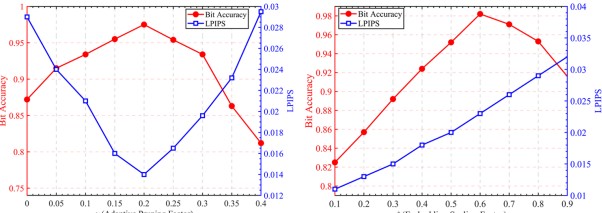

Figure 7: The impact of 3D Gaussians removal is based on the contribution of rendering quality. Declining 3D Gaussians refers to reducing the number of Gaussian primitives. The results are shown for 32-bit messages.

# E  SCALABILITY

Mark3DGS is designed with inherent versatility, enabling its adaptation across a wide range of 3DGS variants beyond its primary implementation on the original 3DGS framework. To evaluate this adaptability, we tested Mark3DGS on four representative extensions: 2DGS Huang et al. (2024a), 4DGS Li et al. (2024a), CompactGS Deng et al. (2024), and DreamGaussian Tang et al. (2023), with results detailed in TABLE 4 and visualized in Fig. 8. Our method consistently delivers excellent visual quality, achieving PSNR values exceeding 23 dB and maintaining low LPIPS scores below 0.270 across all variants, ensuring real-time performance with frame rates above 120 FPS. Additionally, it sustains high watermark extraction accuracy and robust resilience to distortions, underscoring its broad applicability.

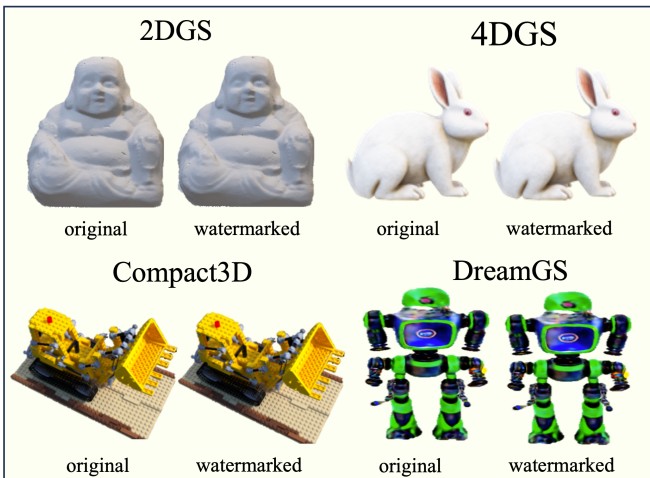

Figure 8: Visual comparisons across 3DGS variants before and after 32-bit watermark message embedding. We display the original image on the left and the watermarked rendering using Mark3DGS on the right for each variant. Despite the structural differences in scene representation, our method successfully embeds watermarks without introducing noticeable visual artifacts.

Specifically, for 2DGS, which lacks the depth of SH coefficients, Mark3DGS leverages structural similarities—such as opacity, position, and color—encoded within semantic codebooks through vector quantization, enabling effective watermark embedding despite the reduced dimensionality. In the dynamic context of 4DGS, which incorporates time-dependent Gaussian attributes, watermarks are embedded in temporally stable components, ensuring reliable extraction even under varying motion conditions. For CompactGS, where pruning reduces model complexity, watermarking is applied post-pruning, achieving an impressive 95.48% extraction accuracy despite significant compression. Finally, in the generative framework of DreamGS, Mark3DGS integrates seamlessly during the synthesis process, preserving both the quality of generated content and watermark robustness through fine-tuning. These results collectively demonstrate our method's robust adaptability and scalability,

making it a versatile solution for protecting intellectual property across diverse 3DGS frameworks in real-world scenarios.

## F  LIMITATION

The Mark3DGS framework, while offering memory-efficient and robust watermarking for 3D Gaussian Splatting models, faces limitations in generalizing pruning across diverse scenes, vulnerability to extreme fine-tuning attacks with full prior knowledge, inefficiencies in tile-based rasterization for variable-density scenes, and high computational demands of the MarkGS-Sim simulator for large-scale scenarios. Future work could enhance adaptability through dynamic pruning, strengthen adversarial robustness with advanced training, optimize tile sizing for efficiency, and develop lightweight simulation for edge devices, broadening its applicability in real-world forensic and security contexts.

## LLM USAGE STATEMENT

In preparing this manuscript, the authors utilized OpenAI ChatGPT-5 to refine the text's clarity and readability. The large language model did not contribute to the generation of technical content, research concepts, or experimental outcomes. After its application, the authors meticulously reviewed and edited all outputs, assuming full responsibility for the final publication.

