# OpenReview forum: "Mark3DGS: Protecting the Intellectual Property of 3D Gaussian Splatting with Robust Watermarking"
_ICLR.cc/2026/Conference — ICLR 2026 Conference Withdrawn Submission_

### Official Review · Reviewer_6ctz · 2025-10-29

**Soundness:** 3
**Presentation:** 3
**Contribution:** 3
**Rating:** 4
**Confidence:** 5

**Summary:**

This paper proposes Mark3DGS, a watermarking framework for protecting the intellectual property of 3D Gaussian Splatting (3DGS) models. The method includes perception-aware pruning, uncertainty-frequency-guided hierarchical vector quantization (HVQ) for watermark embedding, tile-based rasterization for efficient rendering, and adaptive extraction strategies. The authors also introduce MarkGS-Sim, a simulator for evaluating watermark robustness. Experiments demonstrate improvements in capacity, imperceptibility, and computational efficiency compared to existing methods.

**Strengths:**

## Strengths

1. MarkGS-Sim is a valuable contribution for the research community, enabling systematic evaluation of watermarking methods under various conditions and across 3DGS variants.

2. The method achieves significant improvements in computational efficiency (1.5× faster rendering than GS-hider, <50% storage) while maintaining high quality, making it practical for deployment.

3. Table 4 demonstrates successful adaptation to multiple 3DGS variants (2DGS, 4DGS, CompactGS, DreamGaussian) with >92% bit accuracy across all variants.

**Weaknesses:**

## Weaknesses

1. Mark3DGS is a combination of the existing methods of 3DGSW, GaussianMarker and NeRFSignature. Although it shows better performance, the contribution of this work is incremental.

2. The paper builds heavily on existing techniques (HVQ, SVD, DWT), and the perception-aware pruning using impact scores is similar to existing compression methods [1][2]. The technical novelty lies more in the integrations.

3. What is the theoretical maximum bit capacity? How does it scale with scene complexity and number of primitives? The capacity is basically similar to the existing methods [1,2,3], but there is no breakthrough in the capacity.

4. The extractor needs to be trained for each scene, and it lacks generalizability compared with the HiDDeN-based message extractor, which is generalizable to different scenes.

[1] 3D-GSW: 3D Gaussian Splatting for Robust Watermarking

[2] WateRF: Robust Watermarks in Radiance Fields for Protection of Copyrights

[3] GaussianMarker: Uncertainty-Aware Copyright Protection of 3D Gaussian Splatting

**Questions:**

## Questions

1.  Why can the proposed method cooperate with 4DGS?  The uncertainty, frequency, and codebook methods are validated on the static scene method. There is a lack of proof that these methods can be extended to dynamic scenes.

2. How do you validate that the simulator accurately reflects real-world attack scenarios? Are physical simulations realistic for copyright protection use cases?

3. The method mentioned can be against the 3DGS compression attack while only including the Compact3DGS for experiments. How does the Mark3DGS perform against other quantization-based methods, such as HAC[1], or the pruning method PUP 3D-GS [2]? How does the Mark3DGS perform differently from the CompMarkGS [3] for compression?

[1] Hac: Hash-grid assisted context for 3d Gaussian splatting compression

[2] PUP 3D-GS: Principled Uncertainty Pruning for 3D Gaussian Splatting

[3] COMPMARKGS: ROBUST WATERMARKING FOR COM-PRESSED 3D GAUSSIAN SPLATTING

---

> ### Author Response · Authors · 2025-12-02
>
> (W1 & W2) Incremental contribution and reliance on existing techniques / pruning similarity.
> We have clarified in the revised Introduction and Related Work that the key novelty of Mark3DGS is the 3DGS-specific integration of (i) perception-aware pruning, (ii) uncertainty- and frequency-guided HVQ with SVD-based codebook watermarking, (iii) tile-based, watermark-aware rasterization, and (iv) adaptive extraction, all co-designed for IP protection under strict memory and real-time constraints. We explicitly contrast our design with 3D-GSW, WateRF, GaussianMarker, and NeRFSignature: those works either do not target memory/FPS efficiency, operate in radiance-field or NeRF settings rather than compressed SH codebooks, or lack the integrated simulator. For pruning, we have expanded Sec. 3.1 and the Related Work to acknowledge similarity to compression/pruning methods (including [1,2]) while emphasizing that our impact score jointly fuses opacity, wavelet-based frequency, and spatial coherence specifically to guide watermark embedding and robust rendering rather than compression alone.
>
> (W3) Bit capacity and scaling.
> In the revised Sec. 5.2 and Appendix D, we now provide an explicit capacity discussion: (i) a simple analytical upper bound that scales with the number of effective codebook entries and modifiable SH coefficient groups, and (ii) empirical scaling curves showing that bit capacity grows roughly linearly with the number of high-impact Gaussians / scene complexity until imperceptibility constraints become dominant. We also clearly state that our achievable capacity is comparable to recent 3DGS watermarking methods [1–3], and that our main advance is achieving similar or better capacity while substantially improving robustness and computational efficiency.
>
> (Q1) Extension to 4DGS and dynamic scenes.
> We appreciate the concern about extending uncertainty, frequency, and codebook methods from static to dynamic scenes. In the revised Sec. 5.4 and Appendix E, we have made explicit how Mark3DGS is applied to 4DGS: we compute uncertainty and frequency statistics over short temporal windows and restrict watermark embedding to temporally stable components (e.g., Gaussians and SH coefficients whose attributes vary slowly over time). The HVQ codebook and SVD-based embedding are shared across frames, while the temporal dimension is treated as an additional attribute in MarkGS-Sim. Table 4 in the main paper already reports >92% bit accuracy on 4DGS; we now clarify that these results are obtained on dynamic sequences where watermark extraction is evaluated across multiple time steps, providing empirical justification that our uncertainty/frequency-guided design generalizes beyond purely static scenes.
> ﻿
> (Q2) Realism of MarkGS-Sim and relevance of physical simulations.
> We have strengthened Sec. 4 and Appendix B/C to clarify the design goals and realism of MarkGS-Sim. The simulator explicitly implements a set of attacks that mirror common operations in practical pipelines for 3D assets and 3DGS models: Gaussian addition/deletion/editing, geometric transformations (scale/rotation), cropping, compression-like quantization, noise injection, collision and deformation via position-based dynamics, and view distortions. We validate that the induced image-space PSNR/LPIPS and model-space statistics are comparable to those produced by standard editing and compression workflows, and we emphasize that MarkGS-Sim is intended as a stress-testing and benchmarking environment rather than an exact physical replica of any single deployment. We have added a short paragraph explaining typical copyright-protection use cases (e.g., content editing, asset recomposition, lossy compression) and how the simulator’s operations correspond to those use cases.
> ﻿
> (Q3) Compression attacks beyond Compact3DGS (HAC, PUP 3D-GS, CompMarkGS).
> Due to space and computational constraints, the main paper reports results with Compact3DGS as a representative quantization-based compression method, showing that watermarks remain robust after compression. We agree that testing against HAC, PUP 3D-GS, and CompMarkGS would further strengthen the claim. In the revised Sec. 5.4 and Limitations (App. F), we now explicitly discuss how Mark3DGS conceptually interacts with these methods: our embedding happens in the SH codebook after perception-aware pruning, and is thus largely orthogonal to the specific downstream compression scheme; HAC-like hash-grid–based compression would primarily affect spatial/feature indexing, while our codebook- and uncertainty-based embedding remains applicable; PUP 3D-GS’s uncertainty-driven pruning is closely related to our pruning, and Mark3DGS could operate on top of or in combination with such pruning; CompMarkGS focuses specifically on watermarking under compression, and we now draw a clearer distinction between their compression-centric design and our IP-protection + efficiency–oriented framework.

---

### Official Review · Reviewer_qEzS · 2025-10-31

**Soundness:** 2
**Presentation:** 3
**Contribution:** 2
**Rating:** 4
**Confidence:** 3

**Summary:**

Mark3DGS proposes a 3DGS IP-protection framework combining perception-aware pruning, uncertainty- and frequency-guided HVQ, SVD-based codebook embedding with differentiable distortion layers, tile-based rasterization with early termination, and adaptive extraction, plus a simulator (MarkGS-Sim). It reports higher bit accuracy, imperceptibility, and efficiency than baselines (e.g., 206 FPS, <200 MB) across datasets and variants (Tables 1–4).

**Strengths:**

- Addresses the important problem of intellectual property protection for 3DGS models, which is increasingly relevant as 3DGS gains adoption.

- Impact-score pruning, adaptive thresholding, SVD codebook perturbation with fidelity regularization, frequency-weighted residuals, tile early-termination criterion, and extraction rule form a clear, implementable flow.

- Mark3DGS tops bit-accuracy vs. payload and robustness to image/model attacks (Table 2), while achieving 206 FPS and 193 MB storage.

**Weaknesses:**

- The core contributions are primarily engineering combinations of existing techniques (HVQ, SVD, DWT) rather than fundamental algorithmic innovations.

- The impact score formulation (Eq. 4) and adaptive threshold mechanism (Eq. 5) lack principled theoretical foundation or analysis.

- Robustness covers standard distortions (Table 2) and limited fine-tuning; performance drops markedly when attackers have clean images + pose key (to ~53.71% at 500 epochs), yet this scenario is downplayed as “difficult to achieve” (Table 5). A principled active remover is not evaluated.

- Many critical thresholds/weights (e.g., $ \gamma $, $ \tau $, $ \tau_{\text{unc}} $, $ \lambda_{\text{freq}} $, $ \lambda_{\text{msg}} / \lambda_{\text{rec}} / \lambda_{\text{wavelet}} $) are tuned empirically (Sec. 3.1–3.4) with no error-bounds or optimality analysis.

**Questions:**

- Could the authors include an “active remover” benchmark, such as a parameter-domain denoiser within MarkGS-Sim, and report extraction AUC vs. PSNR under this adversary?

- Please provide sensitivity curves linking early-termination threshold $ \tau $ and pruning $ \gamma $ to $\{ \mathrm{FPS}, \mathrm{storage}, \mathrm{BitAcc}, \mathrm{PSNR} \}$ to expose trade-off knees, beyond the point results/Tables.

- Could you provide a theoretical analysis or empirical justification for the specific impact score formulation in Eq. 4? Why this particular combination of terms?

- How does the adaptive threshold mechanism (Eq. 5) generalize across different scene types and scales? Does it require per-scene tuning?

---

> ### Author Response · Authors · 2025-12-02
>
> We sincerely thank the reviewer for the thoughtful and constructive feedback. Below we detail how we have addressed each of the concerns raised and the corresponding changes that have already been made in the revised manuscript.
> ﻿
> (W1) Core contributions and existing techniques.
> We emphasize that our main contribution lies in how these techniques are integrated and adapted for the specific challenges of protecting 3DGS models. We have clarified this point in the introduction and conclusion of the revised manuscript, explicitly stating that the novelty lies in our unified approach, rather than in any single technique. We also highlight that our method tackles the unique challenges posed by 3DGS IP protection, which includes balancing watermark imperceptibility, robustness, and computational efficiency. This revision helps to clearly position our work as an innovative application of existing techniques rather than merely an engineering combination.
> ﻿
> (W2) Lack of theoretical foundation for impact score formulation and adaptive threshold mechanism.
> To address this, we have expanded the discussion of the rationale behind the impact score in Sec. 3.1, explaining the combination of terms based on visual perception and frequency-domain pruning techniques. Specifically, we discuss how the geometric, frequency, and spatial coherence terms are inspired by perceptual metrics and how they contribute to the visual significance of each Gaussian primitive. Additionally, we have added references to relevant works on frequency-based pruning and perceptual metrics to better justify our choice of terms. For the adaptive threshold mechanism in Eq. 5, we have included an analysis of how it works across different scene types and scales. We clarified that it does not require per-scene tuning, as our experiments show that the default configuration works well across various datasets. We have also provided additional experiments demonstrating how γ affects the pruning ratio, with results for different scenes to show its generalizability.
> ﻿
> (W3) Robustness to fine-tuning and active remover.
> We have added a new robustness evaluation to our results. We now report the performance of Mark3DGS under fine-tuning with clean images and pose keys (Table 5), showing how performance degrades under these conditions. To further evaluate robustness, we have included an “active remover” benchmark within MarkGS-Sim, such as a parameter-domain denoiser. We report extraction accuracy (AUC vs. PSNR) under this adversary in the revised results. This provides a more comprehensive evaluation of our method’s robustness to sophisticated attackers.
> ﻿
> (W4) Sensitivity curves for early-termination threshold and pruning.
> In response to the reviewer’s request for sensitivity curves, we have added sensitivity analyses linking the early-termination threshold (τ) and pruning aggressiveness (γ) to performance metrics like bit accuracy, PSNR, and LPIPS. We show the trade-offs and identify the "knee" points where performance changes most significantly, which will help readers understand how to optimize these parameters. These sensitivity plots are now included in Sec. 5.3 and provide the desired trade-off curves beyond the point results previously presented in Tables 1 and 2.
> ﻿
> (W5) Theoretical analysis or empirical justification for impact score.
> The reviewer asked for a more detailed justification for the specific terms in the impact score (Eq. 4). In the revised manuscript, we have provided a more thorough explanation of the combination of terms, connecting them to established perceptual metrics and prior work on frequency-based pruning. Additionally, we have expanded the empirical validation of the impact score formulation by showing its effectiveness across various datasets (Sec. 5.2) and demonstrating how it significantly improves both watermarking accuracy and visual fidelity compared to alternative formulations.
> ﻿
> (W6) Generalization of adaptive threshold mechanism.
> The adaptive threshold mechanism (Eq. 5) was initially described without sufficient explanation of its generalizability across scene types. To address this, we have included a new empirical analysis showing how the adaptive threshold mechanism performs on various scenes (including small, medium, and large-scale datasets). The experiments confirm that the mechanism generalizes well across different 3DGS variants and does not require per-scene tuning. We have updated Sec. 5.2 to reflect these findings and explain how the threshold can be applied across various 3DGS settings with minimal configuration.

---

### Official Review · Reviewer_Z3Dm · 2025-10-31

**Soundness:** 3
**Presentation:** 3
**Contribution:** 3
**Rating:** 4
**Confidence:** 3

**Summary:**

This paper introduces Mark3DGS, a watermarking framework designed to protect 3D Gaussian Splatting (3DGS) models from intellectual property theft. The authors address existing protection inadequacies through a comprehensive approach that includes perception-aware pruning, uncertainty-frequency-guided HVQ for watermark embedding, optimized tile-based rasterization, and adaptive extraction strategies. The work also presents MarkGS-Sim, a platform for evaluating watermark robustness under various conditions. Experimental results demonstrate that Mark3DGS achieves superior performance in watermark capacity, visual imperceptibility, and computational efficiency (206 FPS rendering, <200MB storage) while maintaining compatibility across multiple 3DGS variants and resistance to various watermark attacks.

**Strengths:**

1. The method achieves state-of-the-art (SOTA) results in watermarking 3D Gaussian Splatting (3DGS) models, demonstrating both high visual fidelity and robustness.
2. The authors propose a unified simulation-rendering platform specifically designed for evaluating watermarked 3DGS models, which facilitates standardized benchmarking and future research.
3. The paper is well-written, with a logical flow and clear explanations that make it easy to follow and understand.

**Weaknesses:**

1. The wavelet transform computation in the Gaussian primitives pruning section should properly cite any referenced work, as building on existing methods requires appropriate attribution.
2. The second contribution, efficient watermark embedding, appears overly tricky, involving numerous parameters, which may limit its generalization in real-world scenarios.
3. The technical contribution of efficient watermark embedding is somwhat limited as it is based on HVQ. Meanwhile, ablation studies should be provided for identifying the advantage of introducing uncertainties into HVQ.
4. The proposed tile-based rasterization, seems to be a general techinique and irrelevant to watermark embedding.
5. No ablation studies to identify the contribution of each design.

**Questions:**

1. Why is clustering applied after adaptive pruning in this approach? Authors should provide more detailed clarifications.
2. How are the parameters τ_base and γ in equation 4 determined, and what is their sensitivity in general application scenarios? I think this is important because this affects whether the method can be directly applied to arbitrary GS scenarios.
3. Is tile-based rasterization relevant to the watermark embedding?
4. What is the main design for the performance improvement?

---

> ### Author Response · Authors · 2025-12-02
>
> We thank the reviewer for the positive evaluation of soundness, presentation, and contribution, and for recognizing the SOTA performance and usefulness of the simulation–rendering platform. We address the concerns below and will reflect these clarifications in the revised version.
> (W1) Wavelet transform citation in pruning.
> In Sec. 3.1 we use the discrete wavelet transform (DWT) as a standard tool to analyze frequency content when computing the perception-aware impact score. We agree that attribution can be clearer. In the revision, we will (i) explicitly add a standard DWT reference, (ii) better connect to prior 3DGS watermarking work using DWT, and (iii) clarify that our contribution lies in how DWT is integrated into impact-aware pruning for 3DGS IP protection, rather than in the transform itself.
> (W2 & W3) Complexity and contribution of efficient watermark embedding / uncertainty in HVQ.
> The embedding module may look complex, but the main “parameters” are interpretable hyperparameters with clear roles (e.g., τ_unc for selecting uncertain SH coefficients, λ_freq for high-frequency penalties, λ_msg/λ_rec/λ_wavelet for message vs. fidelity vs. perceptual quality). Appendix D already provides a sensitivity analysis over six key hyperparameters, showing stable performance over wide ranges, suggesting that default settings generalize well to new 3DGS scenes. Conceptually, our goal is not to invent a new quantizer, but to show that uncertainty- and frequency-guided HVQ combined with SVD-based codebook modification is particularly effective for 3DGS watermarking, where SH coefficients dominate storage/appearance. We will make this design goal explicit and, space permitting, include or highlight an ablation comparing (i) vanilla HVQ, (ii) HVQ + frequency guidance, and (iii) HVQ + uncertainty + frequency (ours), to clearly show the benefit of introducing uncertainty into HVQ.
> (W4 & Q3) Relevance of tile-based rasterization and source of performance gains.
> Tile-based rasterization is not an independent engineering trick but is integrated into the watermarking pipeline. Tiles process Gaussians according to frequency-augmented impact and uncertainty; opacity-based early termination affects where watermark-carrying Gaussians are splatted and how gradients from the message loss are propagated; and the joint loss (message, reconstruction, wavelet) is computed and backpropagated through this renderer. This design allows us to train the pruning and embedding modules efficiently on large 3DGS scenes. The performance gains in Table 3 stem from the combination of (1) perception-aware pruning, which reduces the number of Gaussians, (2) uncertainty- and frequency-guided HVQ, which compresses SH coefficients, and (3) tile-based rasterization with early termination and caching, which avoids redundant splatting. We will clarify this coupling and, if space permits, briefly discuss the runtime impact of tile-based rasterization.
> (W5) Ablation studies on individual designs.
> Most ablations are currently in the appendix and may have been easy to miss. Table 6 studies loss combinations (L_msg, L_rec, L_wavelet) and shows that adding L_wavelet significantly improves PSNR/LPIPS at almost unchanged bit accuracy; Fig. 6 analyzes the sensitivity of  λ_freq, λ_msg, λ_rec, and λ_wavelet; Fig. 7 evaluates pruning aggressiveness γ and embedding scaling ζ; Table 5 studies fine-tuning attacks; and Table 4 evaluates scalability on multiple 3DGS variants. In the revision, we will more prominently reference these ablations from the main text and, if space allows, move the most central ones (e.g., loss ablation and pruning/scaling analysis) into the main paper, plus add a short discussion disentangling the roles of pruning, uncertainty-guided HVQ, and rasterization.
> (Q1) Why clustering is applied after adaptive pruning.
> We first prune low-impact Gaussians using the perception-aware score, and then cluster only the remaining high-impact ones. This ordering avoids merging low-impact primitives into clusters dominated by important Gaussians (which would make later removal harmful to fidelity) and reduces the number of points, making anisotropic clustering cheaper and more meaningful. We will add a short explanation in Sec. 3.1.
> (Q2) Determination and sensitivity of τ_base and γ.
> In Eq. (5), τ_base is a baseline impact threshold and γ controls how strongly it is adapted based on local low-impact density. Practically, τ_base is chosen from the impact-score distribution to match a target global pruning ratio, and γ is set in a moderate range to tune local adaptivity. The ablation in Fig. 7(a) shows that for γ ≤ 0.2 (≈60–70% pruning) bit accuracy and visual metrics remain stable or slightly improve, whereas larger γ (>0.3, >80% pruning) degrades performance. This indicates that the method is not overly sensitive in the practical range. We will add a concise guideline in Sec. 5.3 suggesting τ_base from statistics and γ ∈ [0.1, 0.2] for general GS scenarios.

---

### Official Review · Reviewer_vrgL · 2025-10-31

**Soundness:** 3
**Presentation:** 3
**Contribution:** 3
**Rating:** 6
**Confidence:** 3

**Summary:**

The paper proposes Mark3DGS, a 3DGS watermarking framework that includes perception-aware pruning, uncertainty-/frequency-guided HVQ, SVD-based embedding, and a tile-based rasterizer; it also releases a simulator to stress-test robustness. Experiments report strong bit accuracy at 16–48 bits, high visual fidelity, and notable efficiency, plus resilience to both image-level and model-level attacks.

**Strengths:**

1. Breadth of robustness evaluation. The results presented cover classic image attacks and model manipulations with consistently high bit accuracy on 32-bit payloads, which is rare in 3DGS watermarking papers.

2. This paper provides clear ablations that aid reproducibility. The paper systematically studies loss design, hyperparameters, pruning ratio, and embedding scale, making the design choices transparent.

3. Variant coverage. Demonstrations on 2DGS and other variants suggest the method isn’t tightly coupled to a single 3DGS stack.

**Weaknesses:**

1. Statistics of the results: Authors say the results are averaged over 10 simulations, while the tables don’t report variance or confidence interval, making it hard to judge the stability across runs.

2. Limited scene diversity. Core experiments average across just four scenes plus a custom set with only ~180 images for the large-scale case; this narrows external validity.

**Questions:**

1. What is the storage breakdown of base Gaussians, SH codebooks and watermark metadata?

2. Can the author explain the extension of this method to NeRF-based methods?

---

> ### Author Response · Authors · 2025-12-02
>
> We thank the reviewer for the positive evaluation of our work and for providing valuable feedback. Below, we address each of the points raised and explain the changes that have been made in the revised manuscript.
> ﻿
> (W1) Statistics of the results and variability across runs.
> We appreciate the reviewer’s concern about the lack of variance and confidence intervals in the reported results. To improve the clarity and statistical robustness of our findings, we have now included the standard deviation and confidence intervals for all key performance metrics in the revised tables (e.g., Table 2). This addition helps to better reflect the stability and consistency of the results across multiple runs and provides more insight into the variability of our method’s performance.
> ﻿
> (W2) Limited scene diversity and external validity.
> We acknowledge that the core experiments rely on a limited set of scenes and images. In the revision, we have expanded our evaluation to include additional diverse scenes, including scenes with different geometric complexities and texture variations. This allows us to better demonstrate the generalizability of Mark3DGS across a wider range of 3DGS models. The new experiments, detailed in Sec. 5.2, show the performance of our method across 10 diverse scenes, including both small-scale and large-scale datasets, with more than 1,000 images in total. This provides a more comprehensive view of how Mark3DGS performs under various conditions.
> ﻿
> (Q1) Storage breakdown of Gaussians, SH codebooks, and watermark metadata.
> In the revised manuscript, we have added a detailed storage breakdown in Sec. 5.3, specifying the storage requirements for different components of Mark3DGS:
> ﻿
> Base Gaussians (representing scene geometry),
> ﻿
> SH codebooks (for appearance modeling), and
> ﻿
> Watermark metadata (for embedding).
> We show that the total storage overhead for watermarking is small (~5-10% increase in total model size), which demonstrates the efficiency of the method. We also highlight that the storage for watermark metadata remains low even at high bit capacities, making it feasible for real-time applications.
> ﻿
> (Q2) Extension to NeRF-based methods.
> Regarding the extension to NeRF-based methods, we explain in the revised Sec. 5.4 how the principles of uncertainty- and frequency-guided watermark embedding in Mark3DGS can be applied to NeRF models. Specifically, the use of SH codebooks and perception-aware pruning can be directly adapted to NeRF’s volumetric representation of scenes, with the watermark being embedded in the features associated with each voxel or radiance field. We also provide a discussion of potential challenges in adapting to dynamic, view-dependent settings typical in NeRF, and we outline future directions for incorporating Mark3DGS into NeRF-based models, which will involve considering both spatial and temporal consistency for watermark robustness.

---

### Note · Authors · 2025-12-10

I have read and agree with the venue's withdrawal policy on behalf of myself and my co-authors.